# Positioning by Multicell Fingerprinting in Urban NB-IoT Networks [note 1]

**DOI:** 10.3390/s23094266

**Published:** 2023-04-25

**Authors:** Luca De Nardis, Giuseppe Caso, Özgü Alay, Marco Neri, Anna Brunstrom, Maria-Gabriella Di Benedetto

**Affiliations:** 1DIET Department, Sapienza University of Rome, 00184 Rome, Italy; 2Department of Mathematics and Computer Science, Karlstad University, 651 88 Karlstad, Sweden; 3Department of Informatics, University of Oslo, 0373 Oslo, Norway; 4Rohde & Schwarz, 00156 Rome, Italy

**Keywords:** fingerprinting, NB-IoT, positioning

## Abstract

Narrowband Internet of Things (NB-IoT) has quickly become a leading technology in the deployment of IoT systems and services, owing to its appealing features in terms of coverage and energy efficiency, as well as compatibility with existing mobile networks. Increasingly, IoT services and applications require location information to be paired with data collected by devices; NB-IoT still lacks, however, reliable positioning methods. Time-based techniques inherited from long-term evolution (LTE) are not yet widely available in existing networks and are expected to perform poorly on NB-IoT signals due to their narrow bandwidth. This investigation proposes a set of strategies for NB-IoT positioning based on fingerprinting that use coverage and radio information from multiple cells. The proposed strategies were evaluated on two large-scale datasets made available under an open-source license that include experimental data from multiple NB-IoT operators in two large cities: Oslo, Norway, and Rome, Italy. Results showed that the proposed strategies, using a combination of coverage and radio information from multiple cells, outperform current state-of-the-art approaches based on single cell fingerprinting, with a minimum average positioning error of about 20 m when using data for a single operator that was consistent across the two datasets vs. about 70 m for the current state-of-the-art approaches. The combination of data from multiple operators and data smoothing further improved positioning accuracy, leading to a minimum average positioning error below 15 m in both urban environments.

## 1. Introduction

Narrowband Internet of Things (NB-IoT) is a leading technology in the context of low-power wide area networks (LPWANs), standardized by the 3rd Generation Partnership Project (3GPP) in Release 13 (Rel-13, 2016) and aimed at enabling low-cost and power-efficient IoT services over cellular networks [1]. Most of these services either require or benefit from location information [2]. A balance between cost, complexity, energy efficiency, and positioning accuracy is, however, required for massive NB-IoT and calls for NB-IoT-specific positioning technologies, rather than a reliance on global navigation satellite systems (GNSS). A first step in this direction was the integration in Rel-14 (2017) of NB-IoT positioning features inherited from long-term evolution (LTE) that are based on the observed time difference of arrival (OTDOA) technique (see Section 2 for details) [2]. The use of OTDOA in NB-IoT, however, faces several challenges. First, compared with LTE, NB-IoT devices operate on a smaller bandwidth (i.e., between 180 kHz and 200 kHz, depending on the adopted operation mode [3]). Hence, ranging and timing accuracy, on which OTDOA relies, is limited by signal bandwidth. Second, NB-IoT devices are likely to operate in poor coverage scenarios (e.g., deep indoors), and this may negatively affect OTDOA positioning accuracy. Third, the adoption of OTDOA is hindered by the fact that, in most cases, operators have only recently completed or are still performing the deployment of Rel-13 NB-IoT networks [3].

The result is a growing interest in adopting different and novel strategies for NB-IoT positioning. In particular, radio frequency (RF) fingerprinting is an interesting candidate, given its reasonable requirements in terms of device capabilities and network deployment. Indeed, a favorable complexity–accuracy trade-off has made fingerprinting the most popular approach for indoor positioning based on WiFi [4]. Fingerprinting in LTE networks was originally proposed by 3GPP in Technical Specification (TS) 36.809 [5], where RF and timing advance information were combined to define a fingerprint; the approach was however only tested via simulation. As is well known, fingerprinting operates in two phases. During the so-called offline phase, fingerprints, consisting in a set of RF data related to the detected access points of the technology in use (e.g., WiFi Access Points (APs) for WiFi and evolved Node Bs (eNBs) for LTE and/or NB-IoT) are collected at selected positions, referred to as reference points (RPs), and stored in a database. Note that in the case of LTE/NB-IoT, an eNB is associated with multiple physical cell identifiers (PCIs)/narrowband PCIs (NPCIs), each corresponding to a different signal source. Within the subsequent online phase, the location of a target device is estimated as a function of the position of the RPs associated to fingerprints that best match the one provided by the device according to a desired similarity metric, often defined as the inverse of a distance. Among several possible approaches to perform the estimation, *k* nearest neighbors (*k*NN) and its weighted version (W*k*NN) are largely adopted owing to their reasonable complexity–accuracy trade-off [4].

The adoption of fingerprinting in large outdoor areas and using cellular networks poses, however, new, specific challenges both in the offline and online phases. Regarding the offline phase, a high spatial density for RPs and a large number of measurements collected at each RP are typically desirable, as both factors contribute to increasing the positioning accuracy. As a result, the number of RPs defined for the offline phase as well as the number of measurements taken at each RP is typically the result of a trade-off between the time and efforts required to collect the measurements and the desired accuracy. In the case of an outdoor fingerprinting system, given the larger areas to be covered compared to indoor scenarios (up to tens of km^2^ vs. hundreds of m^2^, typically), this trade-off might not be achievable: the need to keep the efforts for the acceptable measurement collection might result in unacceptably low accuracy (Challenge 1). Regarding the online phase, the W*k*NN algorithm is characterized by a computational complexity that grows quadratically with the number of RPs [4]; in considering the coverage area of an outdoor system, the number of RPs might be in the order of thousands or tens of thousands of points. Given the expected high density of devices characterizing the IoT scenarios and the corresponding number of positioning requests, the processing workload during the online phase might be difficult to sustain (Challenge 2). Finally, a challenge common to both phases is related to the density of eNBs/NPCIs available in a commercial deployment: current NB-IoT networks are in fact deployed so as o satisfy coverage requirements defined in terms of communication availability and may not provide enough signal sources to support the accurate positioning by fingerprinting (Challenge 3).

The early attempts to apply RF fingerprinting to NB-IoT did not explicitly assess the above challenges. Fingerprinting in combination with NB-IoT was first investigated in [6], where. however. an indoor scenario similar to the ones typically addressed in WiFi-based fingerprinting was considered. Outdoor fingerprinting in NB-IoT using commercial networks was first analyzed in [7,8], although, due to the hardware limitations of off-the-shelf NB-IoT modems, the RF fingerprint was defined by the received signal strength indicator (RSSI) of only one cell, i.e., the cell to which the NB-IoT device was connected. More recently, fingerprinting using NB-IoT was investigated in [9] using data collected with development boards to avoid the limitations characterizing commercial hardware; however, a limited accuracy was reported. The possibility of adopting fingerprinting in NB-IoT networks as a viable alternative to GNSS positioning remains, therefore, to be fully explored, and the feasibility of this approach in a realistic scenario is yet to be assessed.

A step forward in this direction was taken in [10] by proposing novel design solutions that take into account the information available for NB-IoT signals. This work extends [10] by providing an in-depth analysis of the feasibility and the performance of fingerprinting in NB-IoT networks. The main contributions of this paper can be summarized as follows:It investigates the feasibility of RF fingerprinting in NB-IoT networks and assesses its accuracy in two urban scenarios by leveraging two large-scale measurement campaigns executed in the cities of Oslo, Norway, in 2019 [3] and Rome, Italy in 2021;It proposes several strategies for designing the W*k*NN-based online phase that combines network coverage and RF signal information;It assesses the performance of the above strategies by taking into account (a) the RF parameter adopted in the definition of fingerprint, (b) the amount of RPs collected in the offline phase, and (c) the number of cells considered in the fingerprinting system;It studies the impact on positioning accuracy of combining data from multiple operators and data preprocessing to remove fast fading and discusses the adoption of data combination and preprocessing as a means to achieving a favorable trade-off between offline phase efforts and positioning accuracy;It provides a detailed description of the data used in the performance evaluation and information on how to access the datasets through an open-source license.

The investigation was made possible thanks to the nature of the dataset, in which four RF parameters (RSSI, reference signal received power (RSRP), reference signal received quality (RSRQ), and signal to interference plus noise ratio (SINR)) are provided in a large number of locations and for all the cells detected in such locations, thus overcoming the limitations described in [7].

The paper is organized as follows. Section 2 reviews positioning approaches proposed for NB-IoT. Section 3 introduces the proposed fingerprinting system by first reviewing the W*k*NN algorithm and by then describing the proposed strategies. Section 4 provides a detailed description of the measurement setup adopted for the collection of the two datasets used in this work and of the dataset structure. Next, Section 5 presents the results of the performance evaluation of the proposed strategies on the two datasets and introduces two approaches based on the combination of data by multiple operators and on data smoothing to improve accuracy. Section 6 draws conclusions and suggests future research avenues.

## 2. Background and Related Work

In LTE, the OTDOA algorithm is implemented and run at the network side and is based on time of arrival (TOA) estimates performed by the user equipment (UE) on the periodic downlink position reference signal (PRS) [11,12]. Given a set S of *S* eNBs within its range, the UE selects a reference eNB in the set and reports to the network S−1 TDOA measurements, obtained as the difference between the PRS TOA estimates of each of the remaining S−1 eNBs in S and of the reference eNB, and is referred to as the reference signal time difference (RSTD). The OTDOA algorithm has encountered a widespread adoption in LTE networks worldwide, with reported positioning error in the order of 35 m at the 90% percentile [13,14].

Fingerprinting in LTE was proposed by 3GPP [5] and experimentally validated in [15]. Several extensions were then proposed. In particular, fingerprinting was enhanced in [16] owing to the introduction of RSTD information in the definition of the fingerprint that improved the accuracy over OTDOA.

Ever since Rel-14, NB-IoT uses a dedicated signal, called narrowband PRS (NPRS) to compute OTDOA, similarly to LTE. NPRS can be transmitted continuously (while PRS in LTE is limited to up to six consecutive subframes [17]) in order to compensate for the loss of accuracy due to a narrower signal bandwidth than LTE. Despite the introduction of NPRS, TOA accuracy in NB-IoT is expected to be severely degraded compared to LTE [17]. Recent investigations have proposed and analyzed, via simulation, possible enhancements toward improving the TOA estimation accuracy for NB-IoT at the price, however, of a lack of compliance with the standard and increased complexity (e.g., see [18,19,20]).

The first study proposing fingerprinting using NB-IoT signals was presented in [6], where devices transmitting signals compliant to the NB-IoT standard in terms of baseband data content, bandwidth, and carrier frequency were deployed in an indoor location and used to build a fingerprinting system covering the same location. Albeit interesting, this work did not cover the most realistic use case of fingerprinting with NB-IoTl; that is, covering a large outdoor area served by commercial NB-IoT operators. The earliest contribution addressing such a use case was presented in [7], where RSSI measurements defined fingerprints, and a W*k*NN algorithm—to be used in the online phase—was compared against two other methods: (a) a ranging-based algorithm that uses RSSI to estimate distances between UE and eNB, and (b) a proximity algorithm that matches the position of the UE with an estimate of the position of the eNB detected with highest RSSI. Experimental data collected using commercial NB-IoT devices were used to evaluate the performance of the three above algorithms and showed that RSSI-based fingerprinting outperforms both RSSI-based ranging and RSSI-based proximity algorithms. The best results were obtained with the Pearson $\chi^{2}$ coefficient as the similarity metric, with an average positioning error of about 184 m. Due to limitations in the hardware used to collect data, the fingerprints defined in [7] only included the RSSI of the cell to which the NB-IoT device was connected and thus only included information on a single cell. However, because of cell (re)selection purposes, the NB-IoT standard foresees that a device may collect data on multiple cells at the same time, similarly to what is commonly done by WiFi and LTE devices [1]. In areas characterized by high spatial network density, this may result in a large number of data elements being stored in each fingerprint. Since the amount of information in a fingerprint is a key element in determining positioning accuracy [21], the introduction of multiple cell (in the following: multicell) information in the definition of fingerprints is expected to strongly improve performance with respect to [7].

A first attempt in this direction, beyond the preliminary version of the work presented in [10], was carried out in [9], where Arduino development boards connected to uBLOX N211 NB-IoT modems were used to collect data not only on the serving cell but also on neighboring cells. Collected data for each NPCI detected were collected over an area of $4.5\;{\mathrm{km}}^{2}$ but averaged over “pixels” of $20\times 20\;m^{2}$, leading thus to a spatial discretization of data. WkNN-based fingerprinting was then adopted, using RSRP as RF parameter, and the authors reported an accuracy of about 90 m in the best case.

It is worth noting that NB-IoT is not the only option for positioning using a narrowband technology: recent investigations have focused on other technologies, such as LoRa, with a positioning error between 10 and 20 m [22,23], and SigFox, albeit with rather large errors in the order of several hundreds of meters [24,25]. NB-IoT has, however, the notable advantage of being deployed by mobile network operators, allowing thus to leverage existing infrastructure to position devices, without the need to place anchor nodes to cover the area of interest.

## 3. Proposed Fingerprinting System

This section describes the proposed NB-IoT multicell fingerprinting system. Section 3.1 reviews the W*k*NN algorithm adopted in the online phase for position estimation, while Section 3.2 introduces the proposed fingerprinting strategies to be adopted for evaluating how well RP fingerprints match with the fingerprint of the target device.

### 3.1. The W*k*NN Algorithm

Let *N* be the total number of RPs in the service area A of the positioning system, with $s_{i}$ (i=1,2,⋯,N) being the fingerprint collected at a generic *i*-th RP and with $s_{j}$ being the fingerprint provided by a target device *j*. The position estimation of *j* via W*k*NN relies on the computation of distances $d\left(i,j\right)=d\left(s_{i},s_{j}\right)$ for all RPs, which is followed by the sorting of RPs in terms of increasing distance [26]; the distances proposed in this work are introduced in Section 3.2. Once the k≤N RPs closer to $s_{j}$ according to the adopted distance are selected, W*k*NN provides an estimate ${\hat{p}}_{j}=\left({\hat{x}}_{j},{\hat{y}}_{j},{\hat{z}}_{j}\right)$ of the real position of *j*, denoted as $p_{j}=\left(x_{j},y_{j},z_{j}\right)$ and as follows:(1)$${\hat{p}}_{j}=\frac{\sum_{n=1}^{k}\left(w(n,j)\right)p_{n}}{\sum_{n=1}^{k}w\left(n,j\right)},$$
where $p_{n}=\left(x_{n},y_{n},z_{n}\right)$ is the position of the *n*-th selected RP (n=1,2,⋯,k) in a 3D coordinate system, and w(n,j) is the weight attributed to the *n*-th RP in estimating the position of *j*. In general, the weight w(n,j) can be defined independently from the distance metric d(n,j) [27]; the most common approach, also adopted in this work, is however to choose w(n,j)≡1/d(n,j).

A critical aspect in fine-tuning the W*k*NN algorithm is selecting *k*, both of which are static *k* strategies, where *k* is set once and for all during system setup [28], and dynamic *k* strategies, where *k* is selected at each positioning attempt as a function of $s_{j}$ [29,30], have been proposed in the literature. In this work, a static *k* strategy is adopted, and the impact of *k* on positioning accuracy is investigated.

### 3.2. Proposed Fingerprinting Strategies

The four strategies introduced below provide the definition of distance d(i,j) and a criterion to break ties in the distances associated with two or more RPs during the sorting procedure. As anticipated in Section 3.1, the distance is also used to define the weight of the coordinates of the *k* selected RPs in (Equation 1), i.e., w(i,j)=1/d(i,j). The four strategies explore different ways to combine two pieces of information in defining d(i,j): the number of cells detected in both positions $p_{i}$ and $p_{j}$, and the similarity between the sets of measurements for the selected RF parameter collected in the two positions.

#### 3.2.1. Net

The *Net* strategy uses the number of cells that two locations have in common, without taking into account any RF parameter, as the similarity metric between the two locations. Subsequently, NB-IoT cells are denoted as NPCIs in order to comply with NB-IoT standard terminology. The distance between the *i*-th RP and the location of a target device *j* to be estimated, i.e., the *j*-th test point (TP), is thus defined as follows:(2)$$d_{Net}\left(i,j\right)=\frac{1}{L_{i,j}},$$
where $L_{i,j}\leq L_{j}$ is the number of NPCIs in common between RP *i* and TP *j*, and $L_{j}$ is the number of NPCIs detected in TP *j*. Given the definition in (Equation 2), the distance takes values in the discrete set $\left\{d_{1}=1/L_{j},\;d_{2}=1/\left(L_{j}-1\right),\cdots ,\;d_{L_{j}}=1\right\}$; as a result, sorting RPs based on increasing distance from TP *j* divides RPs in $L_{j}$ groups associated with the distances in the set. Let $G_{m}$ indicate the group of RPs at distance $d_{m}$$\left(m=1,\cdots ,L_{j}\right)$, which is defined as follows:(3)$$G_{m}=\left\{i|d_{Net}\left(i,j\right)=d_{m}\right\}\;\;m=1,\cdots ,L_{j},$$
with $N_{m}=\left|G_{m}\right|\geq 0$ being the corresponding group cardinality. The *k* selected RPs for estimating the position of TP *j* thus include elements of the first *q* groups, with *q* such that
(4)$$\sum_{m=1}^{q}N_{m}≜N_{g}\leq k<\sum_{m=1}^{q+1}N_{m},$$
and $k-N_{g}$ RPs randomly selected within the group $G_{q+1}$.

#### 3.2.2. Enhanced Net

The *Enhanced Net* strategy shares with the *Net* strategy the metric defined in (Equation 2) and the sorting/grouping procedure described in Section 3.2.1. In addition, for each RP *i*, the following distance from the TP *j* is evaluated:(5)$$d_{Enh\_Net}\left(i,j\right)=\sqrt{\sum_{l=1}^{L_{i,j}}{\left(p_{l}^{i}-p_{l}^{j}\right)}^{2}},$$
where $p_{l}^{i}$ and $p_{l}^{j}$ indicate the readings of the selected RF parameter (one among RSSI, RSRP, RSRQ, and SINR) in RP *i* and TP *j* for the *l*-th NPCI in common. RPs within each group are then sorted as a function of increasing $d_{Enh\_Net}\left(i,j\right)$ and in cases where $N_{g}<k$, the first $k-N_{g}$ RPs in the sorted list for group $G_{q+1}$ are selected.

#### 3.2.3. Coverage

The *Coverage* strategy only relies on the selected RF parameter to define the distance metric between RP *i* and TP *j* as follows:(6)$$d_{Cov}\left(i,j\right)=\sqrt{\sum_{l=1}^{N}{\left(p_{l}^{i}-p_{l}^{j}\right)}^{2}},$$
where *N* is the size of the union of all NPCIs detected in any RP and TP and is thus independent from *i* and *j*; whenever $p_{l}^{i}$ and/or $p_{l}^{j}$ are missing, a default baseline value $p_{miss}$ is used for data padding. Note that the value of $p_{miss}$ should be lower than the minimum value present in the data. Based on the range observed for the RF indicators, $p_{miss}=-160$ dBm is used for RSSI and RSRP, and $p_{miss}=-40$ dB is used for SINR and RSRQ. In this case as well, RPs are sorted based on increasing $d_{Cov}\left(i,j\right)$, and the first *k* is selected for estimating the position of TP *j*.

#### 3.2.4. Weighted Coverage

The *Weighted Coverage* strategy explicitly merges network availability and RF information by combining the metrics defined in (Equation 2) and (Equation 6):(7)$$\begin{array}{cc} d_{W\_Cov}\left(i,j\right) & =d_{Net}\left(i,j\right)\cdot d_{Cov}\left(i,j\right)\\ \; & =\frac{1}{L_{i,j}}\sqrt{\sum_{l=1}^{N}{\left(p_{l}^{i}-p_{l}^{j}\right)}^{2}}. \end{array}$$RPs are sorted based on increasing $d_{W\_Cov}\left(i,j\right)$, and the first *k* is selected for estimating the position of TP *j*.

## 4. Dataset Description and Analysis

The experimental data used in this work are part of two large-scale measurement campaigns that took place in the cities of Oslo, Norway, in the summer of 2019 and Rome, Italy, in January 2021. Data were collected using the same measurement setup described in Section 4.1 and were stored using the same structure in described in Section 4.2, that also provides information on how to obtain the data. A comparison between the two datasets is provided in Section 4.3; this is followed by a description of the preprocessing strategies adopted to compensate for their differences in Section 4.4.

### 4.1. Measurement Setup

Measurements were collected using the Rohde & Schwarz (R&S) TSMA6 mobile network scanner [31]. The R&S TSMA6 network scanner is an integrated system comprising an RF receiver, for passive monitoring of downlink control signals transmitted by 3GPP access technologies, and a global positioning system (GPS) receiver, which allows for the association of a geographic information to each measurement. The TSMA6 also includes a Windows-based embedded PC running the ROMES software, used to control and configure the RF and GPS receivers. In addition to the internal RF receiver for passive data acquisition, the TSMA6 can be connected to external devices to collect active data for one or more 3GPP technologies, which are also controlled by the ROMES software. In the Oslo campaign the setup included an Exelonix NB-IoT USB device for the collection of power consumption and active data for NB-IoT [32], while in the Rome campaign, this device was replaced with a Samsung S20 5G mobile phone, which was used to collect 5G active data. The setup used in the Rome campaign is presented in Figure 1 and is drawn from [33]. In this work, only passive data collected by the built-in RF receiver were used. The TSMA6 was calibrated at the Rohde & Schwarz premises before each measurement campaign as part of its periodic calibration schedule. In addition, the TSMA6 features an advanced autocalibration algorithm, which was independently verified to be highly accurate by multiple research groups [34,35]. Additional information on the measurement campaigns and on both passive and active data availability can be found in [3,36,37] for the Oslo campaign and [33,38,39,40] for the Rome campaign.

### 4.2. Data Description and Availability

Each entry in the datasets collected in Rome and Oslo includes an extensive set of data fields, divided into four categories:Time stamp, including date, time, and coordinated universal time (UTC);GPS data, including estimates of latitude, longitude, altitude, speed and heading, and the number of satellites used for the estimation;Network data, including the E-UTRA absolute radio frequency channel number (EARFCN), the corresponding carrier frequency, the NPCI, the mobile country code (MCC), the mobile network code (MNC), the tracking area code (TAC), the cell identity (CI), the eNodeB-ID, and the cell ID.Signal data, including the RSSI, SINR, RSRP, and RSRQ values measured for the reference signals transmitted by the eNodeB on each of the two antenna ports used in NB-IoT (Tx0 and Tx1), as well as the power, RSSI, and carrier-to-interference noise ratio (CINR) values measured on the narrowband secondary synchronization signal (NSSS).

Note that the list of fields for each category provided above is not exhaustive; more details can be found in the description of the dataset in [41].

Table 1 lists the data fields, indicating the corresponding measurement units and formats, and provides an example value.

Not all available data fields were used in this work; the subset adopted includes the latitude and longitude from GPS data, the MNC, NPCI and the eNodeB-ID from the network data, and the RSSI, SINR, RSRP, and RSRQ values averaged over the two antenna ports Tx0 and Tx1 from the signal data. The combined use of NPCI, eNodeB-ID, and MNC was required since the NPCI is not a unique indicator, neither across operators nor within the network of a single operator, due to spatial reuse. Since both datasets include data from multiple operators and cover extensive areas, each set of the RSSI, SINR, RSRP, and RSRQ values in a location was associated with the corresponding <NPCI, eNodeB-ID, MNC> triplet in order to avoid any ambiguity. A triplet of <NPCI, eNodeB-ID, MNC> is indicated in the following as *unique* NPCI or simply NPCI. Data points were then defined as the collection of all data entries collected at the same <latitude, longitude> coordinates, leading to the data point structure shown in Table 2, where NPCI indicates a unique NPCI as previously defined.

Both raw data and processed data, organized in data points as defined above, were released under an open source license in a public repository for both the Oslo and Rome datasets; in addition to the data described in this section, the repository also includes an estimation of the positions of eNodeBs as well as data related to channel impulse responses collected simultaneously with the data described above, opening the means to additional investigations for interested researchers [41].

### 4.3. Datasets Analysis and Comparison

The characteristics of the two datasets collected in Oslo and Rome are as follows:The Oslo dataset includes data from 7 driving measurement runs for a total of 118,880 data entries collected in 5266 data points over an area of approximately 2×2 ${\mathrm{km}}^{2}$. Two different network operators were identified. The runs are presented in Figure 2a;The Rome dataset includes data from 6 campaigns for a total of 31,599 data entries collected in 2670 data points over an area of 8×2.5${\mathrm{km}}^{2}$. Three different network operators were identified. The runs are presented in Figure 2b.

The measurement runs included in each dataset were selected so to guarantee a reasonable spatial overlapping between them.

An early comparison between the number of data entries and data points highlighted that the Oslo dataset is characterized by a larger average number of entries per data point than is the Rome dataset. Most importantly, the data collected in Rome also often showed missing RF parameters for a given NPCI, whereas this phenomenon was seldom observed in the Oslo data. An example of this difference is presented in Figure 3.

As a result, the average number of unique NPCIs with valid RF parameters per data point was slightly larger in the Oslo dataset than in the Rome dataset (11.34 vs. 10.33) despite the presence of a lower number of network operators (2 vs. 3). The number of unique NPCIs with valid RF data further showed a larger variability in the Rome data, as shown in Figure 4, which presents a box plot representation of the data.

Following a review of the setups used to collect the two datasets, the difference was attributed to the heavy additional processing load introduced in the Rome setup by the 5G data collection possibly leading to a slower decoding of NB-IoT signals and corresponding data recording. As a result, at times, RF values would be estimated only for some of the NPCIs detected at a given location, leading to a partial loss of data. This hypothesis was confirmed by an in-depth analysis of the data that identified the presence of “holes” in the RF data for a given NPCI during a run with an average ratio of RF data loss, defined as the ratio between the number of data points where the NPCI was detected and RF data recorded as well as the total number of data points where the NPCI was detected, in the order of $P_{L}=0.25$, with a clear impact on the spatial coherence of data. An example of such impact is presented in Figure 5, which compares the RSSI data collected over time for a sample NPCI in the Oslo vs. Rome datasets and shows the impact of the data loss in the latter set.

Since it could be expected that the loss of spatial coherence would reflect on positioning accuracy (as confirmed by the analysis presented in Section 5), an interpolation-based strategy to compensate such loss was introduced, as discussed in Section 4.4.

### 4.4. Preprocessing of the Rome Dataset

The goal of the preprocessing of the Rome dataset was to compensate for the data loss observed on raw data by introducing synthetic data values obtained by interpolating available data. In order to select the most suitable interpolation technique, the Oslo data were used as a reference dataset to evaluate and compare the performance of the following interpolation techniques:Previous: Indicating with *i* the index of a missing sample $x_{i}$, the sample is replaced with the value of the previous available sample, $x_{i}=x_{k}$, where k<i is the index of the last sample available before $x_{i}$;Next: The missing sample $x_{i}$ is replaced with the value of the next available sample, $x_{i}=x_{l}$, where l>i is the index of the first sample available after $x_{i}$;Nearest: The missing sample $x_{i}$ is replaced with the value of the nearest available sample, $x_{i}=x_{j}$, where *j* is determined as the index of the available sample such that $\left|i-j\right|$ is the minimum;Linear: The missing sample $x_{i}$ is replaced as
$$x_{i}=\frac{\left(i-k\right)x_{l}+\left(l-i\right)x_{k}}{l-k},$$
where l>i is the index of the first sample available after $x_{i}$, and k<i is the index of the last sample available before $x_{i}$;Moving average: The missing sample $x_{i}$ is replaced using
$$x_{i}=\frac{1}{w_{a}}\sum_{j=-m_{1}}^{m_{2}}x_{i+j},$$
where $m_{1}\geq 0$, $m_{2}\geq 0$, and $w_{a}=m_{1}+m_{2}+1$ (missing values are skipped in the computation);Moving median: The missing sample $x_{i}$ is replaced with the median value computed over the set composed of the $m_{1}$ samples before $x_{i}$ and the $m_{2}$ samples after it for a total averaging window length $w_{m}=m_{1}+m_{2}+1$ (missing values are skipped in the computation);Spline: The missing sample $x_{i}$ is replaced by performing a piecewise cubic interpolation using the closest neighboring available samples [42,43];PChip: The missing sample $x_{i}$ is replaced by performing a piecewise cubic interpolation that preserves the curve convexity [44];m-Akima: The missing sample $x_{i}$ is estimated using a modified version of the Akima algorithm [45,46], which is also based on a piecewise polynomial interpolation of degree three or less, in this case determining the slope in the interpolated point as the weighted average of the slopes of the neighboring points; the modification to the original algorithm is in the weights given to the slopes, selected so to reduce overshoot in the vicinity of points with a horizontal slope.

The techniques were evaluated and compared on the Oslo dataset by adopting the following procedure:Randomly select a set of N=500 unique NPCIs to be analyzed and perform the subsequent steps for each NPCI in the set;Artificially remove a fraction $P_{L}$ of RF values equal to the average data loss observed in the Rome data—the removal was carried out by extracting a uniform random variable for each data point including the NPCI under analysis and by removing the RF data if the random variable outcome value was larger than $P_{L}$;Interpolate the missing data using each technique;Evaluate and compare the performance of each technique according to two indicators—the mean square error (MSE) and the determination coefficient $R^{2}$.

In order to define the two performance indicators, let us consider a sequence and assume to replace *m* values of the sequence with estimates obtained with the selected interpolation method. Let us indicate with $y_{i}$ the *i*-th original value and with ${\hat{y}}_{i}$ the corresponding interpolated value; furthermore, let us indicate with $\bar{y}$ the average of the original values:(8)$$\bar{y}=\frac{1}{m}\sum_{i=1}^{m}y_{i}.$$Given the above assumptions, MSE and $R^{2}$ for the interpolation technique used to generate the ${\hat{y}}_{i}$ values are defined as follows:(9)$$MSE=\frac{1}{m}\sum_{i=1}^{m}{\left(y_{i}-{\hat{y}}_{i}\right)}^{2},$$
(10)$$R^{2}=1-\frac{SSE_{res}}{SSE_{tot}},$$
where $SSE_{res}=\sum_{i=1}^{m}{\left(y_{i}-{\hat{y}}_{i}\right)}^{2}$ is the residual sum of squares, and $SSE_{tot}=\sum_{i=1}^{m}{\left(y_{i}-\bar{y}\right)}^{2}$ is the total sum of squares.

MSE and $R^{2}$ are correlated since $SSE_{res}=m\cdot MSE$; however, they can provide complementary information on the performance of the interpolation technique since they take values over different ranges [47]: MSE is defined in [0,+∞), where 0 indicates a perfect interpolation, while $R^{2}$ is defined in (−∞,1], where 1 indicates a perfect interpolation. Note that for reasonably good interpolators, $R^{2}$ typically takes non-negative values; a $R^{2}=0$ corresponds, in fact, to an interpolator that replaces each missing value with the average value $\bar{y}$, that can be considered as a baseline benchmark for the other techniques.

The results of the evaluation are presented in Figure 6 and Figure 7, showing a box plot representation of the performance in terms of MSE and $R^{2}$, respectively; each point in the box plot represents the performance for a specific NPCI according to the considered indicator. The results indicate that all interpolators achieved an average performance better than did the baseline benchmark ($R^{2}=0$); nevertheless, the performance gap between the best interpolator and the worst one was not negligible, as highlighted in Figure 8. Figure 6 and Figure 7 show that several interpolators had a good performance, with an average MSE under 10 and an average $R^{2}>0.7$. Among these, the one that consistently provided the best performance for both indicators in terms of average value and of the variability around it was the linear interpolator, which was thus adopted to preprocess the Rome dataset.

## 5. Performance Evaluation

In this section, results of the performance analysis for the fingerprinting system proposed in Section 3 are presented. The proposed system is compared against the fingerprinting approach proposed in [7], referred to in the following as the *Single Cell* strategy. Note that while in the data used in [7], there was no ambiguity on which cell to consider since only one cell was recorded per location, a criterion must be adopted to select a single cell in the data used in this work. The approach adopted in the following is to associate to each location the NPCI characterized by the best value for the considered RF parameter. This choice is justified by the reasonable assumption that, whenever multiple cells are available, a device will select and connect to the one characterized by the most favorable propagation conditions.

The rest of this section is organized as follows. Section 5.1 describes the setup adopted in the experiments and introduces the performance indicators. Next, Section 5.2 presents the results obtained using the Oslo dataset, while Section 5.3 presents the results obtained using the Rome dataset, focusing on the impact of the linear interpolation technique selected on the basis of the analysis presented in Section 4.4. Finally, Section 5.4 proposes two approaches to further improve positioning accuracy.

### 5.1. Experiment Setup and Settings

All results were averaged over 1000 runs. In each run, data points were first randomly shuffled to avoid any bias related to their collection order; each data point was then randomly labeled as RP, with probability $p_{RP}=0.7$ or TP, with probability $p_{TP}=1-p_{RP}=0.3$. The position of each TP was then estimated by applying the selected combination of the strategy and RF parameter, and the positioning error was measured as the distance between estimated and real positions. The error was then averaged over all the TPs and all the runs, leading to the average positioning error, which was adopted as the first performance indicator in the subsequent subsections. The average positioning error was measured for *k* varying in the range [1,40]; the minimum of the average positioning error as a function of *k* is referred to in the following as minimum average positioning error.

### 5.2. Results for the Oslo Dataset

All results presented in Section 5.2.1 and Section 5.2.2 are based on NPCIs for one of the two operators, referred to as Operator 1, assuming that an operator would build the offline database relying exclusively on its own network deployment. The performance obtained in the Operator 1 vs. Operator 2 networks in Oslo is analyzed in Section 5.2.3.

#### 5.2.1. Impact of Strategies and RF Parameters

Table 3 presents the minimum average positioning error and the corresponding *k* value for each combination of strategy and RF parameter; the RF parameter leading to the smallest minimum average error for each strategy is highlighted in bold. All proposed strategies lead to a higher positioning accuracy than does the *single cell* approach. The performance improvement is limited in the case of the *Net* strategy but is high in all multicell strategies using RF information; the best performance is obtained with the *weighted coverage* using the RSRQ, with an error of 19.5 m for k=3.

All multicell strategies using RF information lead to increased performance for an optimal k≤3. This indicates that in most cases, the first RPs selected as nearest neighbors according to the RF-aware metric are also spatially close to the TP. This is not the case for the *Net* strategy which does not use RF information nor for the *Single Cell* strategy, where a single NPCI is used; for these metrics, a greater error in selecting the nearest neighbors is partially compensated for by an increased number of neighbors *k* in the position estimates. Results in Table 3 also show that for the *Coverage* and *Weighted Coverage* strategies, SINR, RSRP, and RSRQ lead to a similar performance, while RSRP leads to the best results for the *Enhanced Net* strategy, and SINR is the best choice for the *Single Cell* strategy. For all strategies, SINR leads to either the best or second best accuracy; given its consistent performance across strategies, it was thus selected as the reference RF parameter for further analysis.

Figure 9 presents the average positioning error as a function of *k* in the range [1,10] for all strategies, using SINR for the strategies taking advantage of RF information. The results show that the *Enhanced Net*, *Coverage*, and *Weighted Coverage* strategies provide consistently better accuracy than did *Net* and *Single Cell* for all *k* values. In general, the combination of the information on the number of common NPCIs and on RF parameters seems to provide the best positioning performance: the *Enhanced Net* strategy outperforms the *Net* strategy, while the introduction of the number of common NPCIs that differentiates the *Coverage* and *Weighted Coverage* strategies leads to a slight performance improvement in the latter one.

#### 5.2.2. Performance under Limited Information

Results presented in Section 5.2.1 were obtained using all the available information in the dataset, both in terms of (a) spatial density, measured by the number of data points, and (b) network deployment, measured by the number of NPCIs detected at each location. In both cases, however, it is worth investigating how positioning performance is affected by reduced available information.

Spatial density. Given the positioning service area, the desired spatial density determines the number of data points to be collected and, in turn, the time and effort spent in data collection during the offline phase: reducing this number may thus improve the positioning system scalability, making it easier to setup and maintain.In order to assess the impact of a decrease in spatial density, additional experiments were performed, where, in each run, a fraction $P_{RP}<1$ of the points originally labeled as RPs were randomly selected and kept in the offline database. Figure 10 shows the minimum average positioning error as a function of $P_{RP}$ for all strategies using SINR for all but *Net*.The results in Figure 10 show that all strategies are affected by a decrease in spatial density but not all to a similar extent. The *Coverage* strategy, in particular, is heavily affected by the reduction of RPs, with the highest positioning error for low $P_{RP}$ values, which questions its robustness. The *Single Cell* approach shows a relatively low error for $P_{RP}=0.01$ on par with the *Weighted Coverage* strategy; this comes, however, at the price of a low percentage of TPs for which the position is estimated at all. As $P_{RP}$ decreases, in fact, there is an increasing probability that no RP has NPCIs in common with a TP, making position estimation impossible. Although this is true for all strategies, *Single Cell* is the most affected, with only 89% of the TPs with a position fix for $P_{RP}=0.01$ vs. 96% or higher for the other strategies.Note that the optimal *k* value leading to the minimum average positioning error will in general depend on $P_{RP}$; the optimal *k* as a function of $P_{RP}$ is presented in Figure 11, highlighting a trend, common to all strategies, to require a higher optimal *k* as the spatial density increases, suggesting that as more RPs become available, strategies benefit from including a larger number of nearest neighbors.Number of detected NPCIs. Devices might be able to only detect and/or report the strongest detected cells based on their radio and processing capabilities [7]. It is therefore relevant to assess the impact of a reduction in the number of detected NPCIs at each location. This only affects fingerprints collected by the target device and has no effect on the offline database built by the operator. Therefore, experiments were carried out that only preserved the $N_{NPCI}$ strongest/best NPCIs in points to be labeled as TPs; no change was made to data points labeled as RPs. Figure 12 shows the minimum average positioning error for all the proposed strategies as a function of $N_{NPCI}$ in the range [1,9].The results show that, for all strategies, the positioning error decreases as the number of NPCIs used for each TP increases; in all cases, the error is close (within <2%) to the one obtained using all NPCIs (see Table 3) if at least 7 best NPCIs are used. This suggests that a trade-off can be found between complexity (i.e., the amount of information to be collected and sent by the target device) and positioning accuracy. The results also show that as in the case of spatial density, the *Coverage* strategy is the one most affected by a reduction in the number of NPCIs, confirming its poor performance under conditions of scarcity of information.The optimal *k* determining the minimum average positioning error varies with $N_{NPCI}$, as already observed for variations of $P_{RP}$; no clear common trend to all strategies was, however, identified for this parameter. Interestingly, the positive impact of the number of NPCIs on positioning accuracy observed in this work was not detected in [9], where using a single NPCI led to the best performance.

#### 5.2.3. Comparison between Operators

A direct comparison of the accuracy of the proposed strategies on different networks can be carried out by using the data for the two operators detected in Oslo, i.e., Operator 1 (considered so far) and Operator 2. Since the data of the two operators were collected at the same locations, any difference in performance can be interpreted as operator-specific. The analysis carried out in [3] indicated that Operator 2 has overall roughly 30% less NB-IoT eNBs than does Operator 1 (146 for Operator 1 vs. 107 for Operator 2). This difference is also reflected in the dataset used in this work: 160 unique NPCIs were detected for Operator 1 vs. 139 for Operator 2, and the percentage of valid data points, defined as points where at least one NPCI was detected, is 92.1% for Operator 1 vs. 86.9% for Operator 2. Conversely, the average number of NPCIs detected in locations where at least one NPCI was detected is slightly larger for Operator 2 (6.6) vs. Operator 1 (6.1). The full data are reported in Table 4.

The analysis for Operator 2 was carried out using the RSRQ parameter and led to the best accuracy for Operator 1, as shown in Table 3. The results are presented in Table 5 and show that the *Enhanced Net*, *Coverage*, and *Weighted Coverage* provide a similar accuracy for the two networks. Results for the *Net* and *Single Cell* strategies show, however, a markedly worse performance when considering Operator 2; since they are the strategies using less information (no RF information for *Net*, only one NPCI for *Single Cell*), these results indicate that strategies that use more information are not only more accurate but also more robust to information loss.

### 5.3. Results for the Rome Dataset

The analysis carried out on the Rome dataset focused on the impact of the data loss observed during the data processing phase. Data were analyzed per operator as conducted for the Oslo dataset in order to identify the operator to be considered in the analysis. The results are presented in Table 6 and show that the three operators are characterized by similar features; Operator 88, characterized by the highest number of unique NPCIs, was selected for an exhaustive analysis of all combinations of strategies and RF parameters using both the original dataset and the one obtained by linear interpolation, following the approach introduced in Section 4.4.

The results are presented in Table 7 and highlight the marked impact of the data loss on positioning accuracy: all combinations of strategies and RF parameters are in fact characterized by a poor performance using the original dataset as compared to the one observed using the Oslo dataset. The linear interpolation dramatically improves positioning accuracy for all strategies with the exception of the *net* one; this is reasonable, since the *Net* strategy does not use RF parameters but only relies on the presence/absence of a given NPCI in a data point. Since the proposed interpolation technique only fills in gaps for RF parameters in existing NPCIs, its application does not affect the information used by the *net* strategy. The results also show that even when using interpolation, the positioning error in the Rome dataset for Operator 88 is larger than that in the Oslo dataset for Operator 1. This gap might, however, be due to the different characteristics of the data collected for the two operators in terms of the number of unique NPCIs and the average number of NPCIs per valid point, with a marked advantage for Operator 1 in Oslo as evident from the comparison of Table 4 and Table 6; further insight on this aspect is provided in Section 5.4.

The performance of all strategies using data for each of the three operators detected in Rome was subsequently compared, with a focus on the RSRP parameter, which led to the best performance for Operator 88. The results are presented in Table 8 and show that Operator 88 is the one that leads to the best performance in most cases, in particular when the original dataset is considered. The linear interpolation leads to a marked performance improvement for all strategies and for all operators, with the exception of the *net* strategy, as already observed for Operator 88. It is also interesting to observe that the interpolation reduces the performance gap between operators, and in particular between Operator 10 and Operator 88, characterized by the same best absolute performance.

### 5.4. Additional Performance Enhancements

The analysis described din the previous sections showed that the positioning accuracy was largely improved by (a) using as much information as possible, as highlighted by the performance loss suffered by strategies using less information and (b) by performing additional preprocessing on the data, as highlighted by the positive effect of data interpolation. Correspondingly, in this section, two approaches to further improve accuracy are introduced.

#### 5.4.1. Increasing Information: Combining Data from Multiple Operators

Although from an operator perspective, it is appealing to rely only on data under the operator’s control, scenarios can be envisioned where either a third party collects data for multiple operators, as the authors did in this work, and uses all the available data to provide a positioning service, or operators decide to share data under a mutual agreement and provide a common service to their customers. As a result, the set of unique NPCIs for each data point will be given by the combination of the NPCIs detected for each operator, typically leading to more information. Table 9 shows the characteristics of both datasets when information from all operators is combined and can be directly compared with Table 4 and Table 6 for the Oslo and Rome datasets, respectively. The comparison highlights that the combined data are characterized by much larger numbers of unique NPCIs and of valid data points, resulting in a 100% validity of data points.

Table 10 compares the minimum average positioning error obtained for a single operator vs. combined operators data using the *weighted coverage* strategy in combination with the RF parameter that led to the best performance in each dataset, which was RSRQ for Oslo and RSRP for Rome. The results show that combining data leads to a marked performance improvement for all datasets. Interestingly, the combined results for the original Rome dataset also show that the combination of data partially compensates for the performance degradation caused by data loss, leading to an error slightly closer to the one observed for single operators when using the interpolated dataset.

#### 5.4.2. Additional Preprocessing: Data Smoothing

The linear interpolation introduced to compensate for the data loss in the Rome dataset replaced each missing sample with a linear combination of the nearest available samples, which is shown in Section 5.3 to have had a beneficial effect on positioning accuracy. We decided thus to investigate whether additional preprocessing based on linear combinations of available samples might further improve performance. A first step in this direction was to extend the analysis on the impact of data preprocessing on the Oslo dataset presented in Section 4.4 in two ways: first, the performance of the linear interpolation technique was studied as a function of the percentage of RF values removed before interpolation; second, the minimum average positioning error as defined in Section 4.1 was used as the performance indicator. The results are presented in Figure 13, showing that when the percentage of RF values was randomly removed from the data and replaced with a linear interpolation of the data collected in the neighboring data points is in the range [5,45], the positioning accuracy stayed close (within 10%) to the one obtained when using the full original data.

This result can be explained by observing that the RF data values are heavily influenced by the time-varying channel conditions due to fast fading; linear interpolation generates synthetic values as a weighted average of real values and as such provides a smoothed version of the data that partially compensates for the loss of information. Based on this observation, a proper smoothing algorithm was introduced based on the so-called 40λ-rule proposed in [48] to remove fast fading from mobile channel data by using a moving average window. After filtering, each RF value at each data point is replaced with an average of the original measurements collected at data points in a range of ±20λ (i.e., ±≈7.5 m @ 800 MHz) from the data point being considered.

Two different approaches to introduce data smoothing were investigated. In the first approach, referred in the following as “RP smoothing”, smoothing was applied only to data contained in reference points after the split of data points into RPs vs. TPs. This approach aims to model a realistic condition where smoothing can be applied to data collected during the offline phase but not during the online phase. In the second approach, referred in the following as “total smoothing”, smoothing was applied to all data before splitting the data points; this approach was considered in order to assess the advantage of smoothing in a best case scenario where fast fading is also mitigated in the online phase.

Results obtained using the two smoothing approaches are presented in Table 11 for both datasets, using the *Weighted Coverage* strategy in combination with RSRP for Rome and RSRQ for Oslo; for the Rome data, smoothing was applied on the interpolated dataset.

A comparison of results obtained using smoothing vs. the results using raw data reported in Table 10 highlights that smoothing leads to a performance improvements in most cases; the only exceptions are for Operators 10 and 88 in the Rome data when the RP smoothing approach was used. This can be explained by observing that the Rome dataset is already the result of a linear interpolation, thus reducing the advantage of performing additional processing by smoothing. Overall, the joint adoption of smoothing and the combination of data from multiple operators leads to a marked reduction of the minimum average positioning error, with errors below 16 m in all cases, as shown in the rightmost column of Table 11.

It is interesting to observe that both techniques proposed in this Section, that is combining data from multiple operators and smoothing data, improve positioning accuracy with respect to the results presented in Table 3 and Table 8 for Oslo and Rome. Combining information from multiple operators provides, however, in most cases, a more noticeable performance improvement than does smoothing. The only exception is total Smoothing for the Oslo dataset, which leads to a positioning error lower than the one obtained by combining operators; in this case, introducing the combination of information does not lead to any additional performance improvement. Two observations emerge from this result: first, that while total smoothing as defined here is not feasible, some form of processing on data provided by devices during the online phase may improve positioning accuracy. Second, a lower floor for positioning accuracy may exist that cannot be overcome; however, further investigation is required to assess this hypothesis.

## 6. Conclusions

This paper presents a set of positioning strategies of fingerprinting in NB-IoT networks. The strategies, based on the W*k*NN algorithm, determine the position of a target device by combining information about network coverage and signal parameters from multiple cells. The use of multiple cells was made possible by the adoption of the Rohde & Schwarz TSMA6 mobile network scanner for data collection. The TSMA6 can in fact simultaneously collect information on all cells detected at a given location, overcoming the limitations of commercial NB-IoT modems, that typically only provide information on the cell that the device is associated to. The performances of the proposed strategies, in combination with the four parameters available for a NB-IoT signal—that is, RSSI, RSRP, RSRQ and SINR—were evaluated on the experimental data collected in two different cities: Oslo, Norway, and Rome, Italy. The proposed strategies were compared against state-of-the-art algorithms that use single cell and multicell information. The results show that the proposed strategies outperform the previous approaches. The impact of both spatial density in the offline fingerprinting database and of network density was evaluated, and new approaches to improve positioning accuracy based on combination of data across operators and data processing were proposed and evaluated.

The results highlighted the fact that the *weighted coverage* strategy, which combines RF information and network availability, is overall the best choice across the four proposed strategies; it combines the high accuracy guaranteed under conditions of complete information by the *Coverage* strategy only using RF information with the robustness shown by the *Net* strategy only using network availability when information is scarce or incomplete.

The results of the performance evaluation outlined in Section 5 can be used to assess how a fingerprinting system designed according to the proposed positioning strategies meets the three challenges identified in Section 1.

Challenge 1—Trade-off between efforts in the offline phase and accuracy: the data used in this work were collected during driving runs, with no run repetitions and without introducing any form of repeated measurements at each point. This is arguably the most straightforward measurement setup in an outdoor scenario and requires an effort comparable to other outdoor data measurement campaigns associated, for example, to services like Google Street View [49] and is thus definitely feasible for a company (e.g., an operator) willing to build its own positioning service. In addition, the performance improvement obtained by introducing interpolation reported in Section 5.3 suggests that the effort can be further reduced by increasing the spatial density using synthetic fingerprints, thus reducing the number of actual fingerprints collected during the runs. The experimental data collected using the above setup led to an average positioning error of about 15 m when adopting the data combination and preprocessing techniques described in Section 5.4, which is comparable to the error obtained when using GNSS receivers.Challenge 2—Computational complexity and processing workload during the online phase: each of the two datasets used in this work led to the definition of a few thousand RPs, as detailed in Section 4.3. Considering, for example, the Oslo dataset, characterized by an average of 3600 RPs, the average time to serve a positioning request was 0.6 ms on a Dell Precision Tower 3640 workstation, equipped with an Intel Core i7-10700k processor and 32 GBs of RAM, allowing for about 1600 requests per second. Assuming the adoption of the same average distance between two RPs characterizing the datasets (about 3 m) on a regular grid without any gap, covering an area of 1 km^2^ would require approximately 110,000 RPs, leading to about 0.8 s per request in the same setup, assuming an increase of the processing time quadratic with the number of RPs as indicated in Section 1. Extending the grid to an entire city might thus lead in theory to millions of RPs, with a potentially high processing load. However, several approaches can be adopted to address this issue: these include, among others, the introduction of a hierarchical approach based on clustering, as discussed for example in [27], the adoption of a machine learning algorithm in place of W*k*NN at the price of some loss of accuracy [7], or the introduction of dimension reduction algorithms such as the principal component analysis before running W*k*NN. Most importantly, it should be noted that although the total number of RPs may become very high for large urban coverage areas, it can be assumed that a device will use its active NB-IoT connection to transfer its fingerprint to the positioning server, automatically restricting the set of relevant RPs to those falling within the coverage area of the serving eNB.Challenge 3—The number of NPCIs per point and impact on positioning accuracy: the results presented in Section 5 show that the density of eNBs and NPCIs in a commercial network deployment is sufficient to achieve a positioning accuracy comparable to the one provided by a GNSS system, in particular when broadcast signals transmitted by multiple operators are combined.

In conclusion, the results presented in this work support the adoption of fingerprinting based on NB-IoT broadcast signals as a viable and feasible solution to provide accurate position information in IoT applications without the need for dedicated positioning hardware.

Several future research directions can be identified based on the results obtained in this investigation. In particular, the results obtained using the total smoothing technique indicate that data processing in the online phase may markedly improve accuracy. While smoothing cannot be easily performed, a similar effect could be obtained by adopting metrics that combine multiple RF parameters in the definition of a fingerprint, performing thus a sort of averaging across parameters rather than in time (proper averaging) or space (smoothing) for a single parameter. Other future work items include the introduction of machine learning algorithms other than W*k*NN based on singular vector machine and neural networks, and the comparison of OTDOA vs. fingerprinting on experimental NB-IoT data.

## Figures and Tables

**Figure 1 sensors-23-04266-f001:**
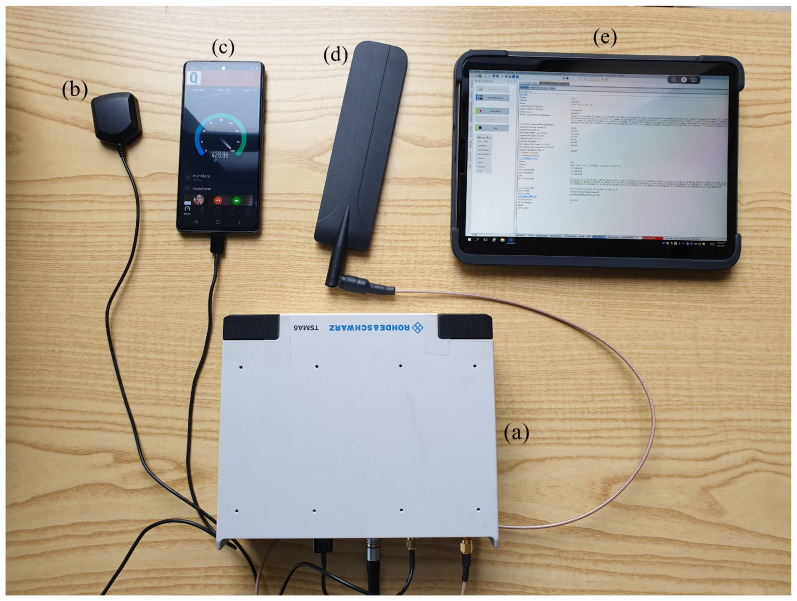
The setup adopted for the measurement campaign in Rome, Italy: R&S TSMA6 mobile network scanner (**a**), GPS antenna (**b**), Samsung S20 5G mobile phone (**c**), RF antenna (**d**), and tablet used to remotely access the ROMES software running in the embedded PC within the TSMA6 scanner (**e**) (image drawn from [33]).

**Figure 2 sensors-23-04266-f002:**
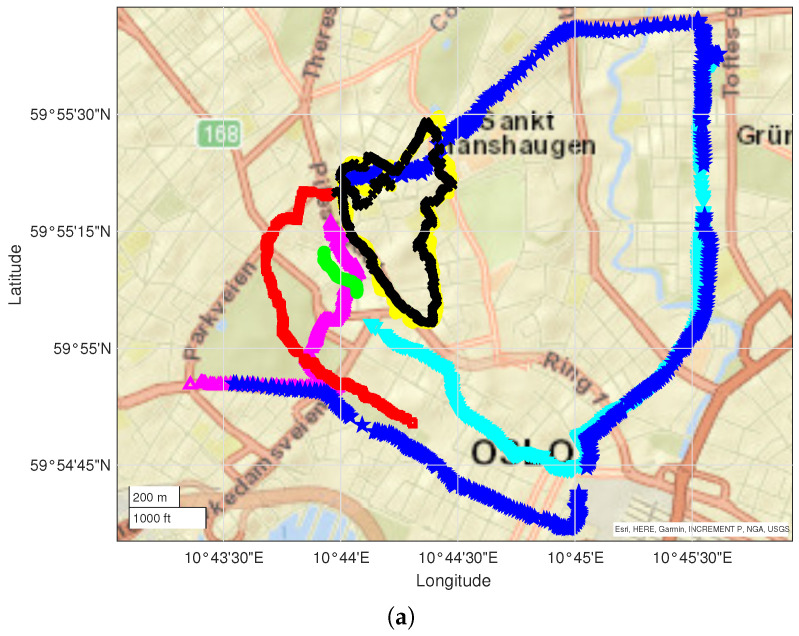
Measurement runs used for the Oslo campaign (**a**) and Rome campaign (**b**); each color represents a different run.

**Figure 3 sensors-23-04266-f003:**
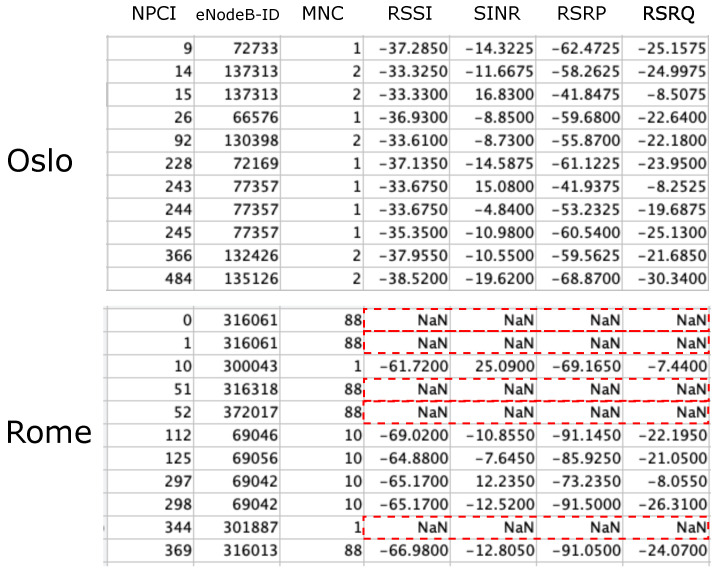
Examples of data points from the Oslo and Rome datasets; missing RF data in the Rome data are highlighted in red.

**Figure 4 sensors-23-04266-f004:**
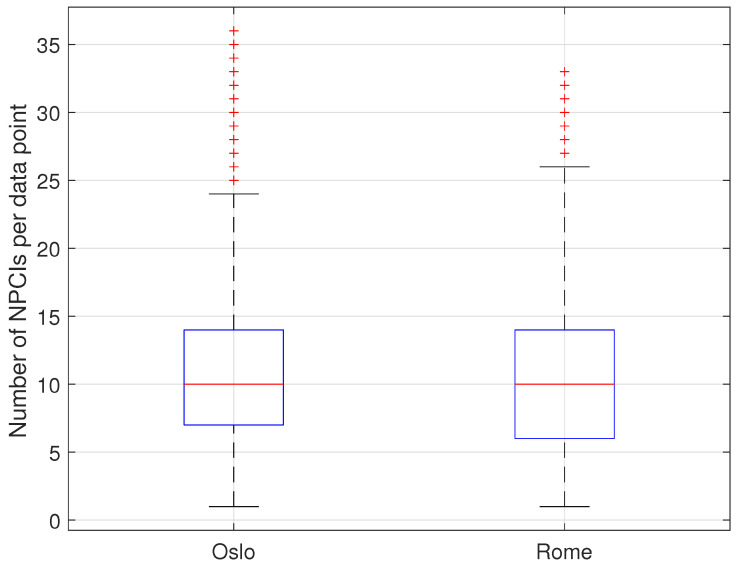
Statistics for the number of unique NPCIs per data point with valid RF data for the Oslo dataset (**left**) vs. the Rome dataset (**right**).

**Figure 5 sensors-23-04266-f005:**
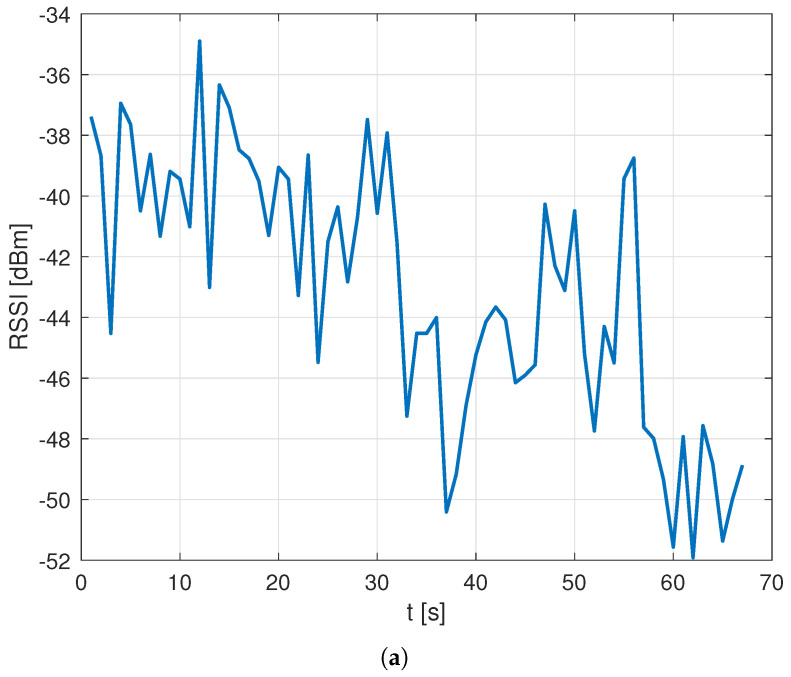
RSSI RF data collected for a sample NPCI as a function of time during a measurement run for the Oslo dataset (**a**) vs. the Rome dataset (**b**).

**Figure 6 sensors-23-04266-f006:**
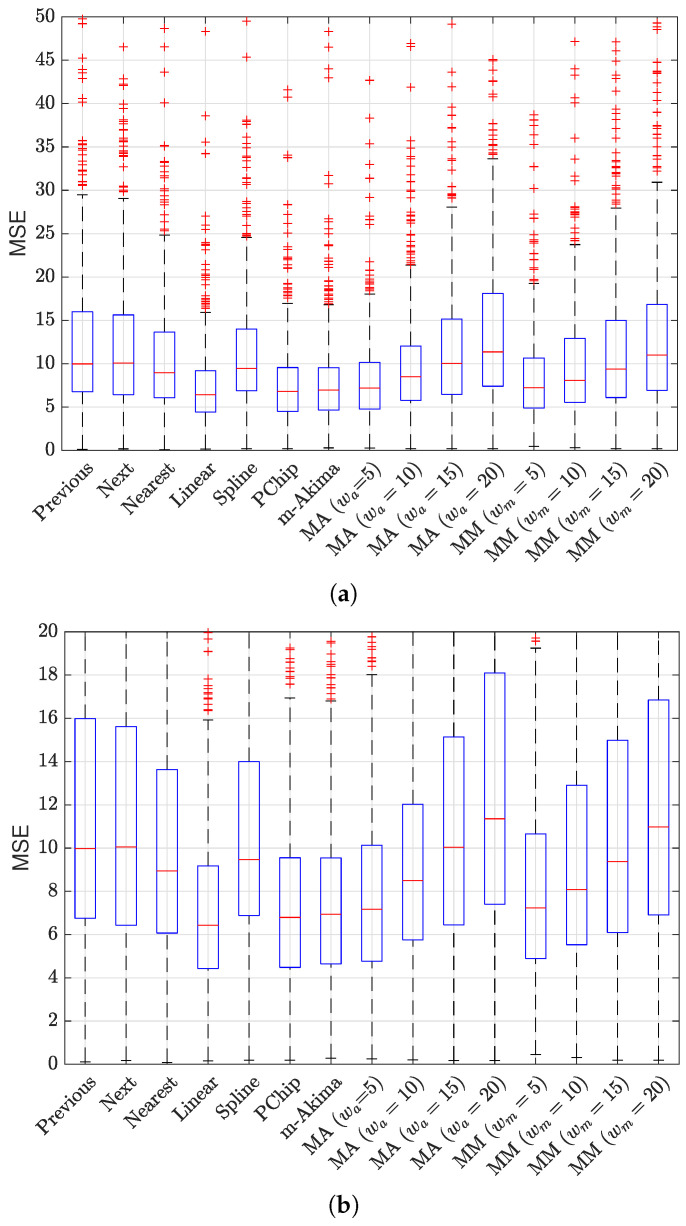
MSE obtained for the considered interpolation techniques (**a**) and the zoom of the same data in the range [0,20] (**b**), where MA stands for moving average and MM for moving median; these techniques were tested for different average window lengths, from 5 to 20.

**Figure 7 sensors-23-04266-f007:**
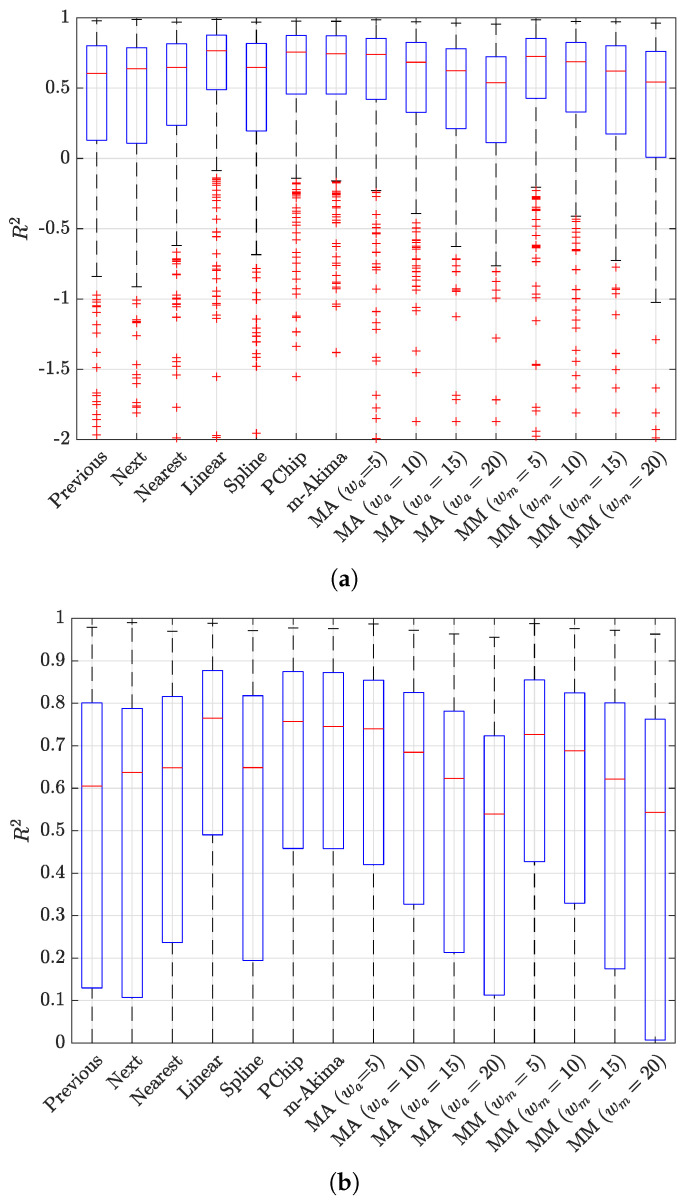
$R^{2}$ obtained for the considered interpolation techniques (**a**) and zoom of the same data in the range [0,1] (**b**), where MA stands for moving average and MM for moving median; these techniques were tested for different average window lengths, from 5 to 20.

**Figure 8 sensors-23-04266-f008:**
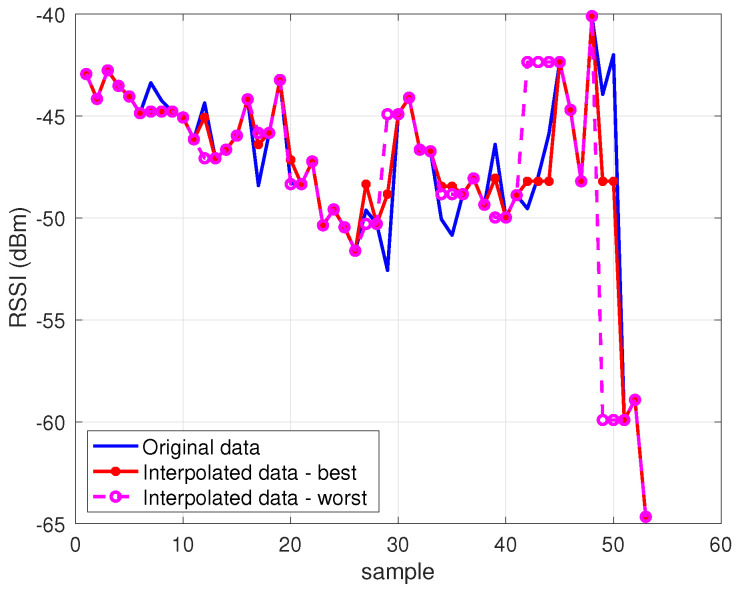
Comparison of best and worst interpolators vs.the original data for a sample NPCI from the Oslo dataset.

**Figure 9 sensors-23-04266-f009:**
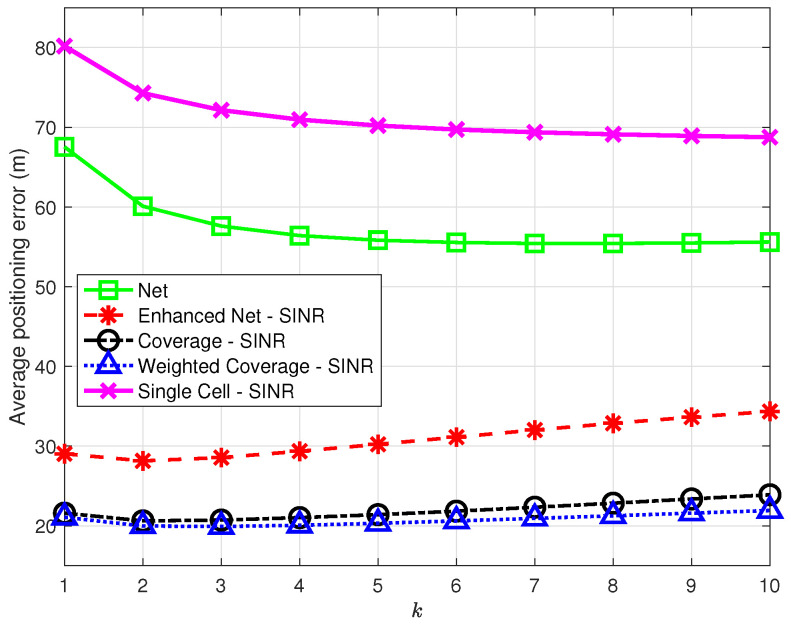
Average positioning error as a function of *k* for the proposed strategies vs. *Single Cell* [7] using SINR in all strategies but *Net*.

**Figure 10 sensors-23-04266-f010:**
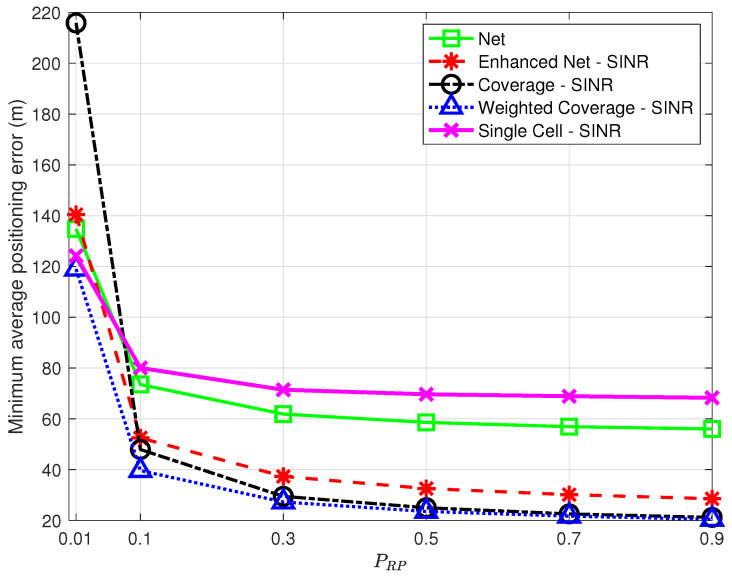
Minimum average positioning error as a function of $P_{RP}$ for the proposed strategies vs. *Single Cell* [7] using SINR in all strategies but *Net*.

**Figure 11 sensors-23-04266-f011:**
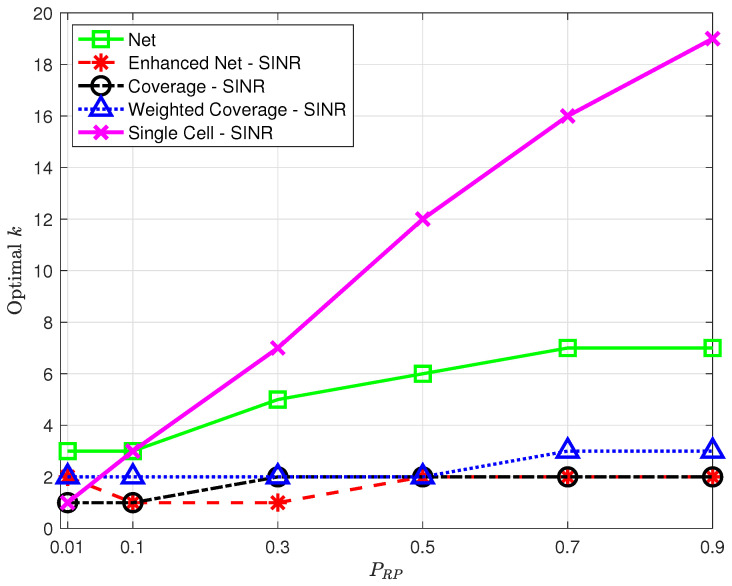
Optimal *k* as a function of $P_{RP}$ for the proposed strategies and the approach proposed in [7] using SINR in all strategies but *Net*.

**Figure 12 sensors-23-04266-f012:**
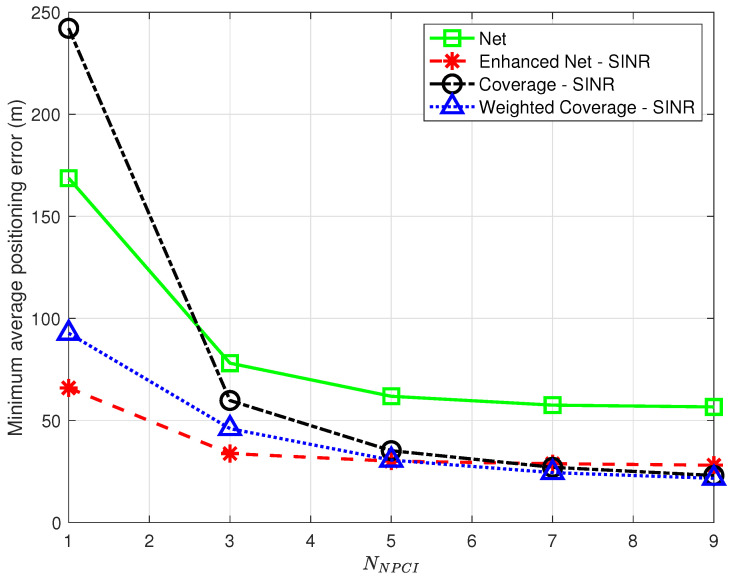
Minimum average positioning error as a function of $N_{NPCI}$ for the proposed strategies using SINR in all strategies but *Net*; the *Single Cell* was not analyzed since, by definition, it operates with $N_{NPCI}=1$.

**Figure 13 sensors-23-04266-f013:**
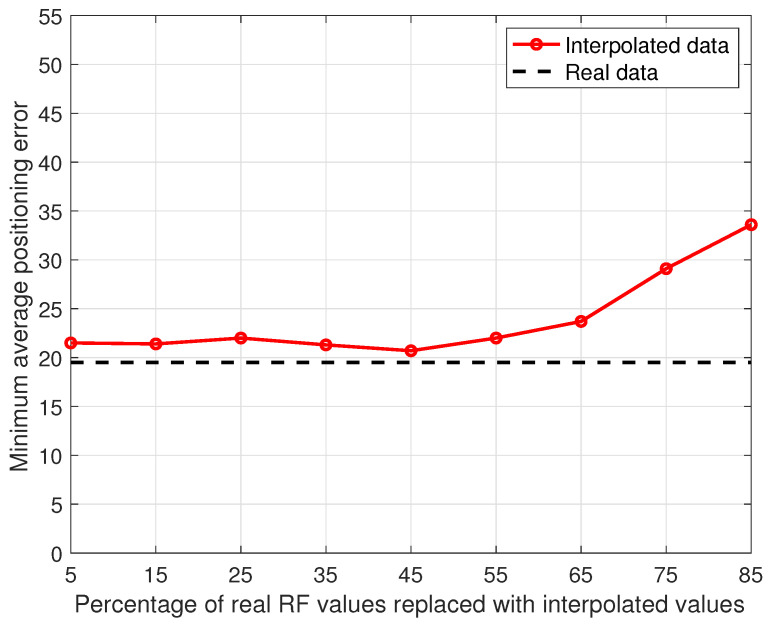
Minimum average positioning error as a function of the percentage of RF values removed and replaced with synthetic values obtained by linear interpolation of remaining values using the Oslo data for Operator 1 and the *weighted average* strategy in combination with RSRQ.

**Table 1 sensors-23-04266-t001:** Data fields included in a data entry and examples of corresponding values.

Category	Data Field	Unit/Format	Example
Time stamp	Date	dd.mm.yyyy	23.07.2019
	Time	hh:mm.s	08:25.5
	UTC	seconds	1,566,583,704
GPS data	Latitude	degrees	59.922214
	Longitude	degrees	10.733242
	Altitude	meters	46.98
	Speed	m/s	8.03
	Heading	degrees	199.36
	Sat	-	6
Network data	EARFCN	-	6352
	Frequency	Hz	811,192,500
	NPCI	-	243
	MCC	-	242
	MNC	-	1
	TAC	-	40,601
	CI	-	19,803,492
	eNodeB-ID	-	77,357
	cell ID	-	100
Signal data	RSSI-Tx0	dBm	−28.4
	RSSI-Tx1	dBm	−28.4
	NSINR-Tx0	dB	20.84
	NSINR-Tx1	dB	26.25
	NRSRP-Tx0	dBm	−40.07
	NRSRP-Tx1	dBm	−34
	NRSRQ-Tx0	dB	−11.65
	NRSRQ-Tx1	dB	−5.59
	NSSS-Power	dBm	−25.3
	NSSS-RSSI	dBm	−25.3
	NSSS-CINR	dB	−25.3

**Table 2 sensors-23-04266-t002:** Information associated with the generic *n*-th data point. The notation NPCI_k,n_ indicates the *unique* NPCI, corresponding to a unique triplet <NPCI, eNodeB-ID, MNC> associated with the *k*-th data entry out of the $L_{n}$ collected at the coordinates <$Latitude_{n},Longitude_{n}$>.

$Latitude_{n}$	$Longitude_{n}$	NPCI_1,n_	RSSI_1,n_	SINR_1,n_	RSRP_1,n_	RSRQ_1,n_
		NPCI_2,n_	RSSI_2,n_	SINR_2,n_	RSRP_2,n_	RSRQ_2,n_
		⋮	⋮	⋮	⋮	⋮
		NPCI_$L_{n},n$_	RSSI_$L_{n},n$_	SINR_$L_{n},n$_	RSRP_$L_{n},n$_	RSRQ_$L_{n},n$_

**Table 3 sensors-23-04266-t003:** Minimum average positioning error and corresponding optimal *k* for the proposed strategies and the *single cell* approach in [7] using the Oslo dataset.

Strategy	RF Parameter	*k*	Error [m]
*Net*	N/A	8	54.0
*Enhanced Net*	RSSI	2	34
	SINR	2	28.1
	**RSRP**	2	**24.0**
	RSRQ	3	30.1
*Coverage*	RSSI	3	23.9
	SINR	2	20.6
	RSRP	2	20.8
	**RSRQ**	2	**20.1**
*Weighted Coverage*	RSSI	3	22.3
	SINR	3	19.9
	RSRP	2	19.9
	**RSRQ**	3	**19.5**
*Single Cell* [7]	RSSI	22	102.4
	**SINR**	19	**66.2**
	RSRP	30	76.9
	RSRQ	30	72.9

**Table 4 sensors-23-04266-t004:** Comparison between data used for the Operator 1 vs. Operator 2 networks in the Oslo dataset; a valid data point is defined as a data point where at least one unique NPCI was detected for the considered operator.

Parameter	Operator 1	Operator 2
Unique NPCIs	160	139
Valid data points	4848	4577
Valid data points (%)	92.1	86.9
Average number of NPCIs per valid data point	6.1	6.6

**Table 5 sensors-23-04266-t005:** Minimum average positioning error and corresponding optimal *k* for the proposed strategies and the approach in [7] using RSRQ as the RF parameter in Operator 1 vs. Operator 2 networks in the Oslo dataset.

Strategy	Operator 1		Operator 2	
	*k*	Error [m]	*k*	Error [m]
*Net*	8	54.0	7	64.2
*Enhanced Net*	2	30.4	2	33.7
*Coverage*	2	20.1	2	19.8
*Weighted Coverage*	3	19.5	2	19.0
*Single Cell* [7]	30	72.9	24	81.9

**Table 6 sensors-23-04266-t006:** Comparison between the data used for the three networks detected in the Rome dataset; a valid data point is defined as a data point where at least one unique NPCI was detected for the considered operator.

Parameter	Operator 1	Operator 10	Operator 88
Unique NPCIs	69	81	87
Valid data points	2670	2669	2670
Valid data points (%)	100	99.9	100
Average number of NPCIs per valid data point	4.9	5.2	4.9

**Table 7 sensors-23-04266-t007:** Minimum average positioning error and corresponding optimal *k* for the proposed strategies and the *Single Cell* approach in [7] using data collected for Operator 88 in the Rome dataset in its original and interpolated versions.

Strategy	RF Parameter	Original		Interpolated	
		*k*	Error [m]	*k*	Error [m]
*Net*	N/A	40 ${}^{★}$	148.4	40 ${}^{★}$	149.1
*Enhanced Net*	RSSI	3	85.3	3	69.5
	SINR	2	63.4	2	39.9
	**RSRP**	2	**55.9**	**1**	**36.5**
	RSRQ	3	64.0	2	43.0
*Coverage*	RSSI	2	76.6	3	36.4
	SINR	2	74.0	2	29.3
	**RSRP**	1	**68.7**	**2**	**26.9**
	RSRQ	2	74.0	2	28.6
*Weighted Coverage*	RSSI	4	57.4	4	35.6
	SINR	3	49.7	2	29.4
	**RSRP**	3	**49.0**	**2**	**26.8**
	RSRQ	3	49.2	2	28.7
*Single Cell* [7]	RSSI	19	213.5	25	215.9
	**SINR**	9	**135.3**	**13**	**117.5**
	RSRP	12	151.5	20	142.9
	RSRQ	10	137.3	17	124.9

${}^{★}$ Although the optimal *k* is at the edge of the considered range, the results show that the error hits a floor for k>40.

**Table 8 sensors-23-04266-t008:** Minimum average positioning error (E) and corresponding optimal *k* for the proposed strategies and the approach in [7] all using RSRP as RF parameter in the three networks detected in Rome using the original and interpolated datasets.

Strategy	Original						Interpolated					
	Op. 1		Op. 10		Op. 88		Op. 1		Op. 10		Op. 88	
	*k*	E [m]	*k*	E [m]	*k*	E [m]	*k*	E [m]	*k*	E [m]	*k*	E [m]
*Net*	25	146.8	32	137.6	40	148.4	35	147.2	32	137.5	40	149.1
*Enhanced Net*	1	82.8	2	62.1	2	55.9	2	47.5	2	33.2	1	36.5
*Coverage*	4	80.1	2	70.9	1	68.7	2	34.8	2	26.7	2	26.9
*Weighted Cov.*	3	69.3	2	58.1	3	49.0	2	35.0	2	26.8	2	26.8
*Single Cell* [7]	32	163.3	20	153.4	9	135.3	40	155.6	25	149.6	13	117.5

**Table 9 sensors-23-04266-t009:** Comparison between the Oslo and Rome datasets using combined data from all operators detected in each dataset; a valid data point is defined as a data point where at least one unique NPCI was detected.

Parameter	Oslo	Rome
Unique NPCIs	299	237
Valid data points	5266	2670
Valid data points (%)	100	100
Average number of NPCIs per valid data point	11.3	15.0

**Table 10 sensors-23-04266-t010:** Minimum average positioning error in meters using each the single-operator and combined-operators data in the Oslo and Rome datasets for the *weighted coverage* strategy in combination with RSRQ (Oslo) and RSRP (Rome); results using both the original and interpolated datasets are presented for Rome.

Dataset	Minimum Average Positioning Error [m]			
Rome (original)	Op. 1	Op. 10	Op. 88	Combined
	69.3	58.1	49.0	46.2
Rome (interpolated)	Op. 1	Op. 10	Op. 88	Combined
	35.0	26.8	26.8	15.3
Oslo	Op. 1	Op. 2	-	Combined
	19.5	19.0	-	16.1

**Table 11 sensors-23-04266-t011:** Minimum average positioning error in meters using the “RP smoothing” and “total smoothing” approaches for the single-operator and combined-operators data in the Oslo and Rome datasets for the *weighted coverage* strategy in combination with RSRQ (Oslo) and RSRP (Rome).

Dataset	Minimum Average Positioning Error [m]			
Rome (RP Smoothing)	Op. 1	Op. 10	Op. 88	Combined
	34.2	26.8	27.1	15.2
Rome (Total Smoothing)	Op. 1	Op. 10	Op. 88	Combined
	25.5	21.9	21.1	14.9
Oslo (RP Smoothing)	Op. 1	Op. 2	-	Combined
	19.0	18.5	-	15.7
Oslo (Total Smoothing)	Op. 1	Op. 2	-	Combined
	13.8	14.5	-	14.5

## Data Availability

Data used in this work are available for download from a public repository hosted on Zenodo [41].

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
