# Peer review of "Positioning by Multicell Fingerprinting in Urban NB-IoT Networks†"

_sensors, 2023, doi:10.3390/s23094266_

Round 1

Reviewer 1 Report

This paper presents and compares a set of positioning strategies for NB-IoT networks using fingerprinting and the WkNN algorithm. The strategies are evaluated on the real-world experimental data collected in two different cities. The authors also discuss the approaches to improve positioning accuracy based on data processing and combination.

Strengths:

1. The paper provides a clear explanation of the proposed positioning strategies and how they work.

2. The paper provides enough details about the experimental setup and data analysis methods to allow for reproducibility.

Weaknesses:

1. Limited novelty. Although the authors perform a lot of detailed analysis on the proposed approaches. Fingerprinting is a common positioning technology to determine a target’s position. KNN has also been widely used by researchers to realize the fingerprinting for decades due to the simplicity. Additionally, the data processing modules, including the interpolation and smoothing, are intuitive and straightforward. Therefore, the paper’s contribution is considered limited and may need more justification.

2. It would be better if the authors can provide a more detailed explanation of the limitations of existing fingerprinting approaches and how the proposed strategies overcome them.

3. Fingerprinting requires a lot of effort on data collection and offline training and cannot be generalized well to the new unseen environment. The practical use of the fingerprinting still needs to be verified.

4. It would be necessary to discuss the computational complexity and scalability to larger networks.

5. The description in line 453-459 on page 16 is not consistent with the Figure 10.

Reviewer 2 Report

The paper shows well organisation and excellent presentation, it is well written too. I have a few remarks: 

1. can you make the figures clearer?

2. The authors did not mention if the equipment they used is calibrated. 

3. Did the authors consider to compare the proposed methods with published papers in literature? 

4. Please arrange the keywords alphabetically 

5. Both MSE and RMSE are used, why MSE is used here instead? 

6. During the collection of data, did you remove the effect of fast fading by taking many measurements over a small area (function of wavelength) and then performing averaging? 

7. A summary table that summarises the pros and cons of investigated strategies will be helpful. 

Round 2

Reviewer 1 Report

The authors have addressed my previous concerns.